# Clinical and genomic landscape of gastric cancer with a mesenchymal phenotype

Sang Cheul Oh[1,2], Bo Hwa Sohn[1,3], Jae-Ho Cheong[4], Sang-Bae Kim[1,3], Jae Eun Lee[4], Ki Cheong Park[4], Sang Ho Lee[5], Jong-Lyul Park[6], Yun-Yong Park[7], Hyun-Sung Lee [1,3], Hee-Jin Jang [1,3], Eun Sung Park[8], Sang-Cheol Kim[9], Jeonghoon Heo[10], In-Sun Chu[11], You-Jin Jang[12], Young-Jae Mok[12], WonKyung Jung[12], Baek-Hui Kim[13], Aeree Kim[13], Jae Yong Cho[14], Jae Yun Lim[14], Yuki Hayashi[15], Shumei Song[15], Elena Elimova[15], Jeannelyn S. Estralla[15], Jeffrey H. Lee[15], Manoop S. Bhutani[15,16], Yiling Lu[1,3], Wenbin Liu[1,3], Jeeyun Lee[17], Won Ki Kang[17], Sung Kim[18], Sung Hoon Noh[4], Gordon B. Mills[1,3], Seon-Young Kim [6], Jaffer A. Ajani[15] & Ju-Seog Lee [1,3]

Gastric cancer is a heterogeneous cancer, making treatment responses difficult to predict. Here we show that we identify two distinct molecular subtypes, mesenchymal phenotype (MP) and epithelial phenotype (EP), by analyzing genomic and proteomic data. Molecularly, MP subtype tumors show high genomic integrity characterized by low mutation rates and microsatellite stability, whereas EP subtype tumors show low genomic integrity. Clinically, the MP subtype is associated with markedly poor survival and resistance to standard chemotherapy, whereas the EP subtype is associated with better survival rates and sensitivity to chemotherapy. Integrative analysis shows that signaling pathways driving epithelial-to-mesenchymal transition and insulin-like growth factor 1 (IGF1)/IGF1 receptor (IGF1R) pathway are highly activated in MP subtype tumors. Importantly, MP subtype cancer cells are more sensitive to inhibition of IGF1/IGF1R pathway than EP subtype. Detailed characterization of these two subtypes could identify novel therapeutic targets and useful biomarkers for prognosis and therapy response.

[1] Department of Systems Biology, The University of Texas MD Anderson Cancer Center, Houston, TX 77030, USA. [2] Department of Internal Medicine, Guro Hospital, College of Medicine, Division of Hemato-Oncology, Korea University, Seoul 08308, Korea. [3] Institute for Personalized Cancer Therapy, The University of Texas MD Anderson Cancer Center, Houston, TX 77030, USA. [4] Department of Surgery, Yonsei University College of Medicine, Seoul 03722, Korea. [5] Department of Surgery, Kosin University, College of Medicine, Busan 49267, Korea. [6] Personalized Genomic Medicine Research Center, Korea Research Institute of Bioscience and Biotechnology, Daejeon 34141, Korea. [7] Department of Medicine, ASAN Institute for Life Sciences, ASAN Medical Center, University of Ulsan College of Medicine, Seoul 05505, Korea. [8] Medical Research Institute, College of Medicine, Inha University, Incheon 22212, Korea. [9] Department of Biomedical Informatics, Center for Genome Science, National Institute of Health, Daejeon 34141, Korea. [10] Department of Molecular Biology and Immunology, Kosin University, College of Medicine, Busan 49267, Korea. [11] Korean Bioinformation Center, Korea Research Institute of Bioscience and Biotechnology, Daejeon 34141, Korea. [12] Department of Surgery, Guro Hospital, College of Medicine, Korea University, Seoul 08308, Korea. [13] Department of Pathology, Guro Hospital, College of Medicine, Korea University, Seoul 08308, Korea. [14] Medical Oncology, Yonsei University College of Medicine, Seoul 03722, Korea. [15] Department of Gastrointestinal Medical Oncology, The University of Texas MD Anderson Cancer Center, Houston, TX 77030, USA. [16] Department of Gastroenterology, Hepatology, and Nutrition, The University of Texas MD Anderson Cancer Center, Houston, TX 77030, USA. [17] Department of Medicine, Samsung Medical Center, Division of Hematology-Oncology, Gangnam-Gu, Seoul 06351, Korea. [18] Department of Surgery, Samsung Medical Center, Gangnam-Gu, Seoul 06351, Korea. These authors contributed equally: Sang Cheul Oh, Bo Hwa Sohn, Jae-Ho Cheong, Sang-Bae Kim. Correspondence and requests for materials should be addressed to J.-S.L. (email: jlee@mdanderson.org)

Gastric cancer is the third leading cause of cancer-related death and the fifth most commonly diagnosed cancer in the world[1]. Although surgery is frequently recommended for localized gastric cancer[2], the benefit from surgery alone is limited to patients with relatively early-stage disease. To prevent recurrence and improve the survival rates of patients after surgery, multimodal therapies, including chemoradiation and chemotherapy, have been established[3]. However, such treatments have improved cure rates by only ~10% and have increased toxicity. Both preoperative chemotherapy and postoperative chemoradiation chemotherapy have shown benefit over surgery alone[4–6]. However, >50% of patients still succumb to their cancer.

Even with the advent of adjunctive therapies, however, the optimal approach for an individual patient or subset of patients is difficult to determine[3]. There is considerable clinicopathologic heterogeneity among tumors and outcomes among patients considered to have similar clinical or pathologic disease remain unpredictable[7,8]. This inherent clinical heterogeneity is consistent with the biological differences among patients with gastric cancer.

During the course of tumor progression, cancer cells with an epithelial origin frequently acquire a mesenchymal phenotype through epithelial-to-mesenchymal transition (EMT), which is a physiologic process in which epithelial cells acquire phenotypic, motile, and invasive characteristics of mesenchymal cells[9,10]. EMT can be mediated by many different signaling pathways, including the transforming growth factor (TGF)-β pathway[11]. Although EMT is considered to be one of the biological features promoting clinical heterogeneity in gastric cancer[12–14], the molecular and genetic characteristics of gastric cancer with an acquired mesenchymal phenotype and the clinicopathologic significance of these characteristics are not fully understood.

In the current study, we used a genome-wide survey of gene expression data to uncover potential subtypes of gastric cancer that have distinct biological characteristics associated with prognosis, as well as signaling pathways enriched in each subtype that could serve as potential therapeutic targets. Here, we report that gastric cancers with a mesenchymal phenotype are associated with poor prognosis, markedly low somatic mutation rates and microsatellite instability (MSI), resistance to standard chemotherapy, and sensitive to inhibition of IGF1/IGF1R pathway.

## Results

**Gene signature reflecting a mesenchymal phenotype.** We first performed unsupervised hierarchical clustering analysis using gene expression data from 93 human gastric cancer tissue samples from a Korea University Guro Hospital (KUGH) cohort (Supplementary Table 1) to uncover potential gastric cancer subtypes. To estimate the degree of heterogeneity among the tumors and between gastric cancer and non-gastric cancer in the stomach, we included in the microarray experiments and analysis three samples of gastrointestinal stromal tumor (GIST), the most common mesenchymal cancer in the gastrointestinal tract[15], that had been located in the stomach. Unsupervised clustering revealed two distinct clusters (Fig. 1a): the small (S) cluster and the large (L) cluster. Interestingly, the three GIST tissue samples were co-clustered with samples in cluster S. The gastric cancer tissue samples in cluster S shared a considerable gene expression patterns with GIST tissue samples, indicating that the samples in cluster S potentially harbored biological characteristics similar to those of mesenchymal cells. Kaplan–Meier plots and log-rank tests indicated that patients whose tumor samples fell into the S cluster had significantly lower overall survival (OS) and recurrence-free survival (RFS) rates than did patients whose tumor samples fell into the L cluster (P = 0.007 for OS and 0.006 for RFS (Supplementary Table 1); Fig. 1b, c).

We next sought to identify genes whose expression was unique to the subtype identified by cluster S, and uncovered 299 genes with differential expression between the two clusters (Supplementary Fig. 1a and Supplementary Data 1). Gene network analysis revealed that the TGF-β pathway was the most significantly activated pathway in tumors in cluster S, as evidenced by increased expression of genes directly regulated by TGFB1, TGFB3, and SMAD3 (Supplementary Table 2 and Supplementary Fig. 2). Moreover, SMAD7, an inhibitor of the TGF-β pathway[16], was significantly inactivated in the cluster S genetic signature, offering further supporting evidence that the TGF-β pathway was activated in tumors in cluster S. The TGF-β pathway is best known for inducing EMT[12], suggesting that tumors in cluster S lost epithelial characteristics and acquired a mesenchymal phenotype. Moreover, gene network analysis revealed that α-catenin, a typical epithelial cell marker in the

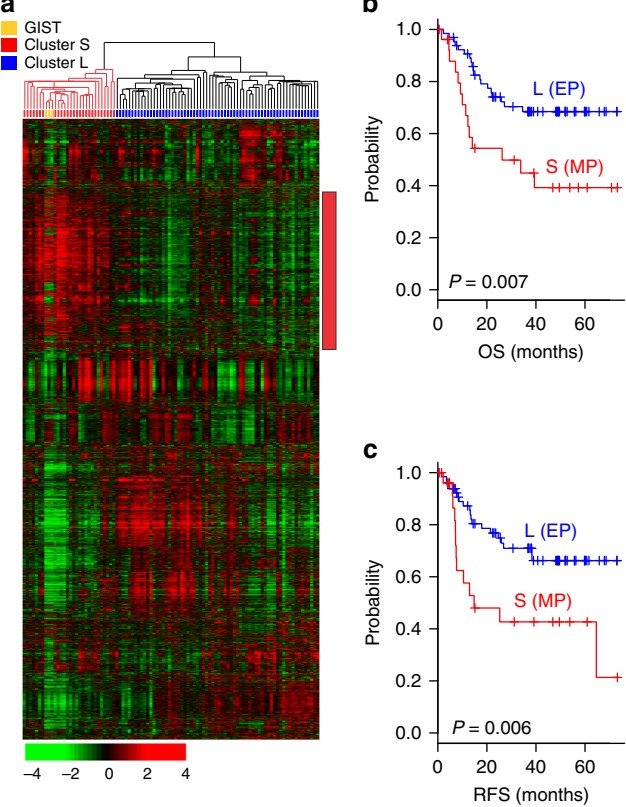

**Fig. 1** Hierarchical clustering analysis of gene expression data from the Korea University Guro Hospital (KUGH) cohort. **a** Hierarchical clustering of gene expression data from 93 patients with gastric cancer and 3 patients with a gastrointestinal stromal tumor (GIST) located in the stomach in the KUGH cohort (n = 93). Genes with an expression level that had at least a twofold difference relative to the median value across tissues in at least 15 tissues were selected for hierarchical clustering analysis (3931 gene features). The data are presented in a matrix format in which each row represents an individual gene and each column represents a tissue sample. Each cell in the matrix represents the expression level of a gene feature in an individual tissue sample. The red and green coloring in the cells reflects relatively high and low expression levels, respectively, as indicated in the scale bar (log₂ transformed scale). **b**, **c** Kaplan–Meier plots of the two gastric cancer clusters in the KUGH cohort (n = 93). The three patients with GIST were excluded for plotting. P values were obtained using the log-rank test. The + symbols indicate censored data. EP epithelial phenotype, MP mesenchymal phenotype, OS overall survival, RFS recurrence-free survival

gut[17], was the most significantly decreased RNA in tumors in cluster S (Supplementary Table 2). Given this finding, together with the similarity of the gene expression pattern in cluster S to that of the mesenchymal GIST samples, we referred to tumors in cluster S as mesenchymal phenotype (MP) and those in cluster L as epithelial phenotype (EP) gastric cancers ("MP subtype" and "EP subtype" hereafter).

**Validation of the MP subtype and its prognostic significance.** After identifying the MP genomic signature reflecting EMT and demonstrating its association with poor prognosis in patients with gastric cancer, we sought to validate the association of the MP subtype with prognosis in independent patient cohorts. For this validation, we used gene expression data from the Yonsei University Severance Hospital (YUSH) cohort (n = 65; Supplementary Table 1 and Fig. 2a). When patients in the YUSH cohort were stratified according to MP or EP subtype by the Bayesian compound covariate predictor (BCCP) algorithm, Kaplan–Meier plots showed significant differences in OS (P = 0.001 by log-rank

test) and RFS (P = 0.003 by log-rank test) rates between patients with the two different subtypes of tumors (Fig. 2b and Supplementary Fig. 3a).

To further test the robustness of the MP genetic signature and prognostic relevance of the two subtypes, we generated gene expression data from another independent cohort (Kosin University College of Medicine (KUCM) cohort, n = 109; Supplementary Table 1). The prognosis for patients with MP subtype tumors was significantly poorer than for those with EP subtype tumors (P < 0.001 by log-rank test for both OS and RFS; Fig. 2c and Supplementary Fig. 3b). When we applied the MP signature in two independent large cohorts of patients (Samsung Medical Center (SMC) cohort, n = 432 and Asian Cancer Research Group (ACRG) cohort, n = 300; Supplementary Table 1) [18,19], we again found that patients with MP subtype tumors had significantly shorter OS and RFS rates than did those with EP subtype tumors (Fig. 2d, e and Supplementary Fig. 3c, d, P < 0.001 for both OS and RFS). We also applied the MP signature in patients with different ethnic backgrounds (white and Hispanic) from a cohort at The University of Texas MD Anderson Cancer

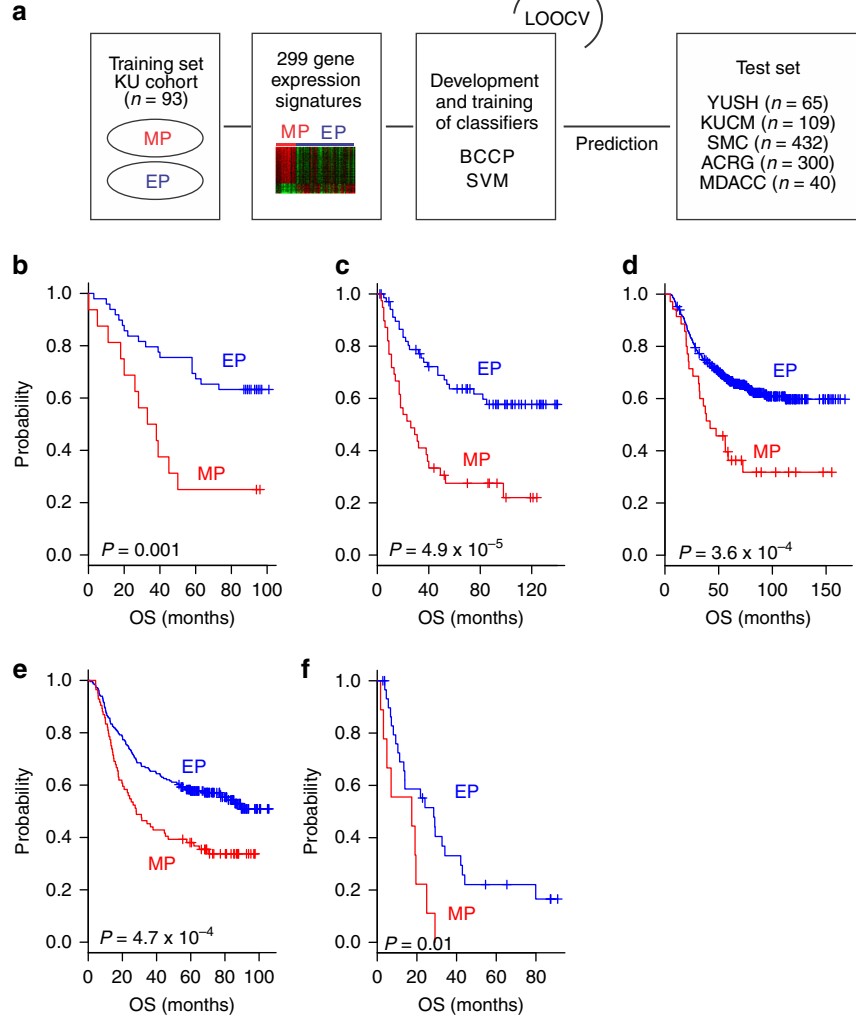

**Fig. 2** Construction of prediction models in the validation cohorts. **a** A schematic overview of the strategy used to construct prediction models and evaluate predicted outcomes on the basis of gene expression signatures. EP epithelial phenotype, MP mesenchymal phenotype, BCCP Bayesian compound covariate predictor, SVM support vector machine, LOOCV leave-one-out cross-validation, YUSH Yonsei University Severance Hospital, KUCM Kosin University College of Medicine, SMC Samsung Cancer Research Institute, ACRG Asian Cancer Research Group, MDACC The University of Texas MD Anderson Cancer Center. **b–f** Kaplan–Meier plots of overall survival (OS) in patients with EP or MP subtype gastric cancer predicted by BCCP in the YUSH cohort (**b**), KUCM cohort (**c**), SMC cohort (**d**), ACRG cohort (**e**), and MDACC cohort (**f**). P values were obtained using the log-rank test. The + symbols in the panels indicate censored data

Center (MDACC cohort; $n = 40$; Supplementary Table 1). Consistent with results from the other cohorts, in which most patients were ethnically Asian, OS rates in patients in the MDACC cohort with MP subtype tumors were significantly lower ($P = 0.01$ by log-rank test) than in those with EP subtype tumors (Fig. 2f). In addition, when a support vector machine (SVM) algorithm was applied to the same gene expression data as an independent prediction model, we observed highly concordant predicted outcomes from SVM and the BCCP as evidenced by a significant difference in prognosis between SVM-predicted subtypes and a significantly high Cohen's kappa coefficient in all tested cohorts (Supplementary Fig. 4). Taken together, these results demonstrated the robustness of MP signature regardless of differences in ethnicity and prediction algorithms.

To evaluate the prognostic value of the MP signature in combination with other clinical variables, we next carried out univariate and multivariate Cox proportional hazards regression analyses with combined clinicopathologic variables in a pooled cohort (KUGH, YUSH, KUCM, SMC, and ACRG cohorts combined; $n = 999$). In addition to depth of tumor invasion, lymph node invasion, distant metastasis, and American Joint Committee on Cancer (AJCC) stage, which are well-known prognostic factors, MP or EP subtype was a significant predictor of RFS in the univariate analysis (Supplementary Table 3). When we included all relevant clinical variables in a multivariate Cox regression analysis, we found that MP or EP subtype remained a significant prognostic factor (MP subtype hazard ratio (HR) 1.7, 95% confidence interval (CI) 1.36–2.2, and $P = 2.7 \times 10^{-6}$ by likelihood ratio test for RFS; Supplementary Table 3).

We then tested the prognostic independence of the MP or EP subtype against current staging systems. When patients with different AJCC disease stages were stratified by tumor subtype, the subtype successfully identified high-risk patients at all AJCC stages (Fig. 3). Taken together, these findings suggest that the MP signature retains its prognostic relevance even after classic clinicopathologic prognostic features have been taken into account.

**Resistance of the MP subtype to chemotherapy**. Because more than half of patients in the KUGH, YUSH, and KUCM cohorts had received adjuvant chemotherapy, we next sought to determine whether each subtype was also associated with a difference in clinical benefit from adjuvant chemotherapy. Because patients with locally advanced gastric cancer have been shown to benefit most from adjuvant chemotherapy, we performed a subset analysis of patients in the KUGH, YUSH, and KUCM cohorts with

AJCC stage II, III, or IV disease without distant metastasis ($n = 180$). Of the 180 patients, 132 received adjuvant chemotherapy. These patients were divided by tumor subtype (MP or EP), and the difference in RFS rates was independently assessed. Adjuvant chemotherapy was associated with significantly increased RFS rates in patients with EP subtype tumors (3-year RFS rate of 70.1% for those who received chemotherapy compared with 43.4% for those who did not; $P = 0.003$ by log-rank test; Fig. 4b). The HR for recurrence among those who received adjuvant chemotherapy was 0.42 (95% CI 0.22–0.8, $P = 0.004$ by likelihood ratio test). However, no benefit from adjuvant chemotherapy was observed among patients with MP subtype tumors (3-year RFS rate of 44.8% for those who received chemotherapy compared with 42.1% for those who did not; $P = 0.98$ by likelihood ratio test; Fig. 4c).

We also carried out an interaction test to assess the true difference (heterogeneous treatment effect) between the two tumor subtypes in terms of the impact of adjuvant chemotherapy. When the Cox regression model was applied, the interaction of the MP and EP subtypes with adjuvant chemotherapy reached significance ($P = 0.01$ by likelihood ratio test; Fig. 4d), demonstrating that patients with EP subtype tumors benefit from adjuvant chemotherapy more than patients with the MP subtype.

**Genomic and proteomic landscape of the two subtypes**. We next investigated the molecular characteristics of the MP and EP subtypes using genomic and proteomic data recently generated from gastric cancer tissues ($n = 262$) from The Cancer Genome Atlas (TCGA) project[20]. Analysis of these data revealed that the somatic mutation rate was significantly lower in MP subtype tumors than in EP subtype tumors ($P = 2.4 \times 10^{-8}$ by Student's $t$-test; Fig. 5), suggesting that the DNA repair system is relatively intact in MP subtype cancer cells. In EP subtype tumors, consistent with a higher mutation rate, *MLH1*, a DNA mismatch repair gene, was significantly silenced through methylation of its promoter regions. In contrast, among MP subtype tumors, with a low mutation rate and intact MLH1 activity, we did not observe the MSI that is typical in colon cancer with mutations in the DNA mismatch repair genes and hypermutations[21]. In addition, the MP subtype was weakly associated with a lack of Epstein–Barr virus (EBV) infection, which is known to be associated with better prognosis[22,23].

We next analyzed proteomic data generated using reverse phase protein arrays from TCGA tumor samples. Consistent with our gene expression data, E-cadherin and α-catenin, two of the most important epithelial adhesion proteins in epithelial cells[17],

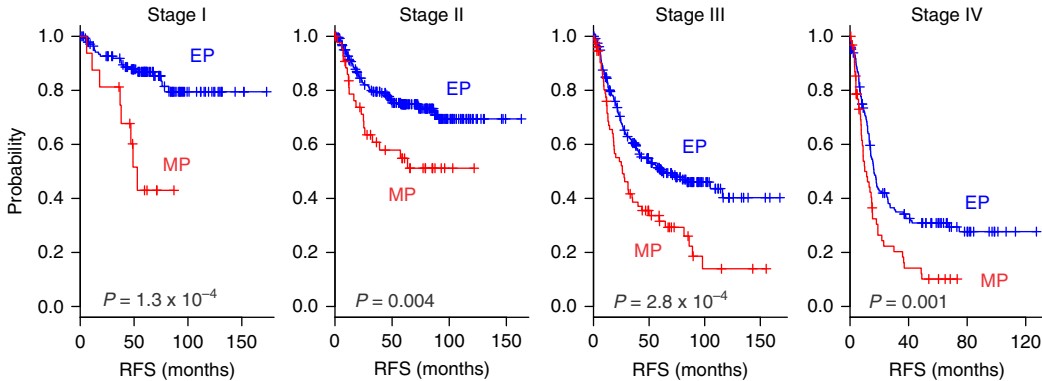

**Fig. 3** Prognostic significance of mesenchymal phenotype (MP) independent of AJCC stages. When patients were stratified according to stages in the pooled cohort (KUGH, YUSH, KUCM, SMC, and ACRG in total of $n = 999$), MP remained associated with poor prognosis regardless of stages. $P$ values were obtained using the log-rank test. The + symbols in the panels indicate censored data. EP epithelial phenotype, MP mesenchymal phenotype, RFS recurrence-free survival

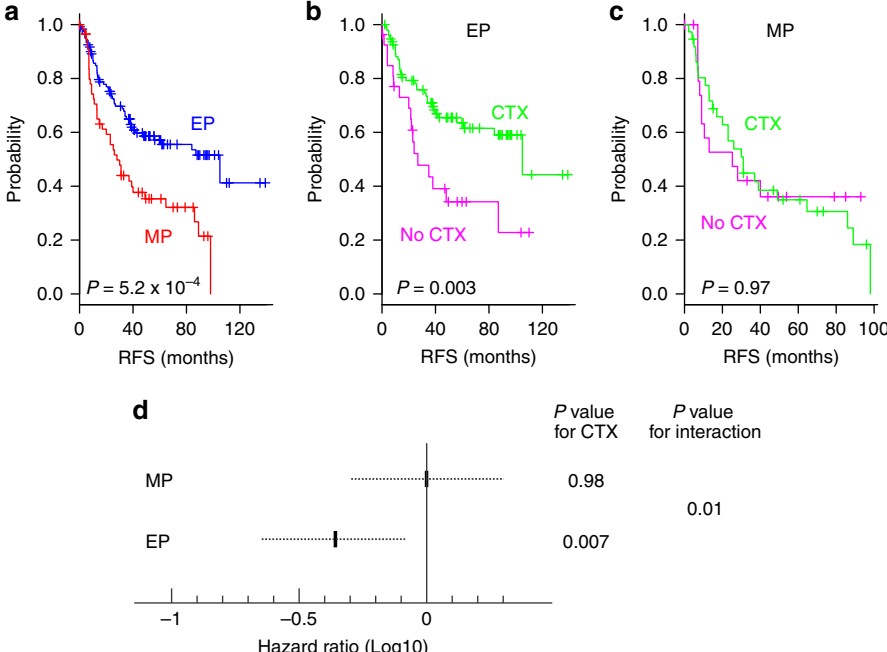

**Fig. 4** Clinical significance of MP subtype. **a–c** Kaplan–Meier plots of recurrence-free survival (RFS) in patients with American Joint Committee on Cancer (AJCC) stage II, III, or IV disease without distant metastasis. Analysis was performed for this subset of patients pooled from three cohorts (Korea University Guro Hospital, Yonsei University Severance Hospital, and Kosin University College of Medicine cohorts; $n = 180$). **a** Kaplan–Meier plots of RFS among patients with each subtype of gastric cancer (EP epithelial subtype, MP mesenchymal subtype). **b**, **c** Kaplan–Meier plots of RFS among patients who received adjuvant chemotherapy (CTX) and those who did not (No CTX) for each tumor subtype. $P$ values were obtained using the log-rank test. **d** Interaction of tumor subtype with adjuvant chemotherapy in patients with gastric cancer. The Cox proportional hazards regression model was used to analyze the interaction between tumor subtype and adjuvant chemotherapy (CTX). The dotted lines represent the 95% confidence intervals of the hazard ratios. EP epithelial phenotype, MP mesenchymal phenotype

were among the most downregulated proteins (Supplementary Fig. 5) in MP subtype tumors, strongly suggesting that MP subtype tumors lose epithelial characteristics. The MP subtype was also associated with high expression of MYH11 (smooth muscle myosin, heavy chain 11), RICTOR, and CAV11 (caveolin 11), which are markers for mesenchymal lineage or are involved in EMT[24–28]. Taken together with our gene expression data, these findings strongly suggest that MP subtype tumors appears to have an increased potential for invasion and metastasis.

We also assessed genome-wide copy number alterations in the two subtypes. Although both subtypes shared highly similar copy number alterations across chromosomes, the magnitude of copy number loss was substantially lower in MP subtype tumors (Supplementary Fig. 6), suggesting that MP subtype tumors have a more stable genome. Analysis of RNA-sequencing data from TCGA tumor samples also uncovered significantly upregulated microRNAs (miRNAs) in MP subtype tumors. miR-490, which is best known for promoting EMT[29], showed the highest level of upregulation among MP subtype tumors (Supplementary Fig. 7), providing further supporting evidence that MP subtype tumors have mesenchymal characteristics.

When we tested for an association between MP or EP subtype and Lauren histologic classification[30], we found that diffuse histologic type was significantly more common among MP subtype tumors (61.9% of MP subtype tumors and 38.1% of EP subtype tumors; $P = 1.5 \times 10^{-7}$ by $\chi^2$ test; Supplementary Table 4), suggesting a potential connection between histologic and molecular subtypes. Consistent with the diffuse type having a higher stromal or non-tumor content, MP subtype has higher non-tumor content than EP subtype (Supplementary Fig. 8a). Likewise, estimation of non-tumor content using CIBERSORT[31]

also showed a higher percentage of non-tumor cells in MP subtype (Supplementary Fig. 8b).

**Biological insights of the two subtypes**. We next identified genes whose expression patterns were specific to the MP subtype and were conserved in gastric cancer tissue samples from five cohorts (KUGH, YUSH, KUCM, TCGA, and ACRG), as described in the Methods. Expression of 605 genes was significantly different between MP and EP subtype tumors in all five cohorts (Fig. 6). Because MP subtype has higher non-tumor content, we estimated contribution of non-tumor cells to gene expression level by removing the proportion of gene expression from non-tumor cells (see Online Methods). When the contribution of non-tumor cells was removed, the difference between MP and EP subtype was not substantially changed (Supplementary Fig. 8c). Likewise, adjustment of gene expression data with CIBERSORT showed similar results (Supplementary Fig. 8d), suggesting that non-tumor or stromal cells were not key contributor of gene expression difference between two subtypes.

When the 605 differentially expressed genes were ranked according to expression ratios between MP and EP subtype tumors, members of the *secreted frizzled-related protein* (SFRP) family, key inhibitors of WNT ligands[32], were among the top-ranked genes. Of the five known members of the *SFRP* family, four *SFRP*s (SFRP1, SFRP2, SFRP3 (also known as FRZB), and SFRP4) were significantly downregulated ($P < 0.001$ by Student's $t$-test and <3-fold) in EP subtype tumors, indicating that the WNT pathway might be highly activated in EP subtype tumors (Supplementary Fig. 9a). Expression of these genes was not significantly different after adjustment for non-tumor cell effects (Supplementary Fig. 10). Downregulation of *SFRP*s was mediated

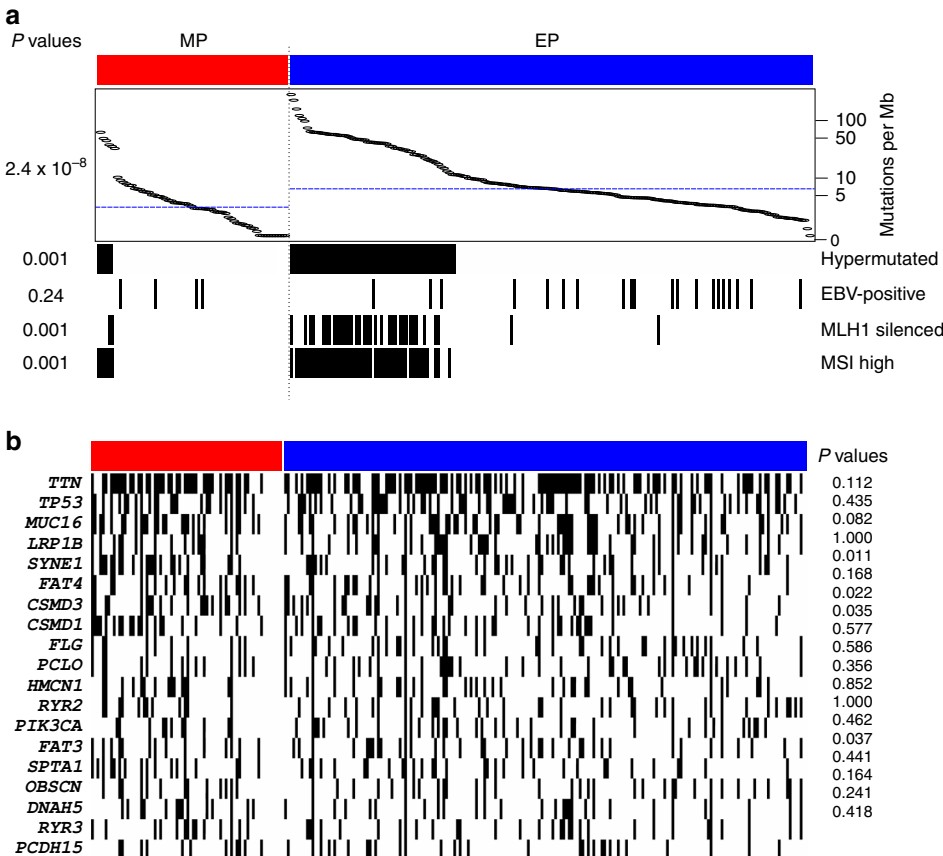

**Fig. 5** Genomic landscape of mesenchymal phenotype (MP) and epithelial phenotype (EP) subtypes of gastric cancer. **a** Genomic and histologic data were retrieved from The Cancer Genome Atlas project database and analyzed ($n = 262$). Tumors with greater than 11.4 mutations/Mb were classified as hypermutated tumors. ALL $P$ values were obtained using $\chi^2$ tests except for mutation rates. $P$ values for mutation rates were obtained using Student's $t$-test (two-sided). **b** Mutational landscape of two subtypes in TCGA cohort. Most frequently mutated genes in TCGA data are presented (20 genes)

by hypermethylation of the promoter regions of these genes, as evidenced by increased methylation in EP subtype tumors (Supplementary Fig. 9b) and significant inverse correlation between promoter methylation and messenger RNA (mRNA) expression of all four *SFRP*s (Supplementary Fig. 11). Down-regulation of SFRPs in EP subtype was further validated in another cohort (YUSH, Supplementary Figs. 12 and 13).

In addition, expression of *GLI* transcription factors (*GLI1, GLI2,* and *GLI3*), which are key transcription factors of the hedgehog pathway[33], was significantly upregulated in MP subtype tumors, suggesting potential activation of the hedgehog pathway in MP subtype tumors. Indeed, gene network analysis revealed that the hedgehog pathway was highly activated in MP subtype tumors (Supplementary Fig. 14). Consistent with previous analysis, gene network analysis of the 605 conserved genes also revealed that the TGF-β pathway was significantly activated in MP subtype tumors (Supplementary Fig. 15).

**IGF1 and IGF1R pathway as therapeutic target for MP subtype.** Most interestingly, expression of *IGF1* was significantly upregulated in MP subtype tumors (Fig. 6) with significance remaining the same even after adjustment of expression data for non-tumor cell effect (Supplementary Fig. 10). To uncover potential mechanisms for upregulation of *IGF1* in MP subtype tumors, we analyzed genomic copy number and promoter methylation of the *IGF1* gene. Of the 70 MP subtype tumor samples from TCGA, 11 showed amplification of the *IGF1* gene, and expression of *IGF1* in these tumors was significantly higher

than in normal tissue and in EP subtype tumors (Fig. 7a, b). The promoter region of *IGF1* was significantly hypomethylated in the remaining MP subtype tumors (Fig. 7a, c). Consistently, mRNA expression of *IGF1* was inversely correlated with promoter methylation (Fig. 7d). Hypomethylation of the IGF1 promoter in MP subtype was further validated by pyrosequencing of promoter region in tumors from the YUSH and KUCM cohorts (Supplementary Fig. 16a). In good agreement with the TCGA data, methylation status was inversely correlated with expression of mRNA (Supplementary Fig. 16b), further supporting methylation-mediated regulation of *IGF1* expression in MP subtype tumors. Taken together, these data strongly suggest that upregulation of *IGF1* in MP subtype tumors might contribute to early recurrence and resistance to chemotherapy.

Since analysis of genomic data from multiple cohorts suggested activation of the IGF1 pathway in MP subtype, we assessed activation status of the IGF1 receptor (IGF1R) in gastric cancer tissues with western blot experiments. In good agreement with our analysis, IGF1R was significantly more activated (phosphorylated) in MP subtype than in EP subtype as evidenced by higher phosphorylation of the receptor in MP ($P = 3.2 \times 10^{-4}$ by Student's $t$-test, Fig. 7e, f). We next tested if MP gastric cancer cells were more sensitive to inhibition of IGF1R. Gastric cancer cell lines were first grouped according to expression of IGF1. Of the six cell lines examined, Hs746T, SNU1, and MKN74 expressed IGF1 (MP-like), while the remainders (MKN28, MKN45, and SNU16) lacked expression of IGF1 (EP-like) (Fig. 8a). In good agreement with this, phosphorylation of IGF1R in cell lines correlated with IGF1 expression (Fig. 8b). Treatment

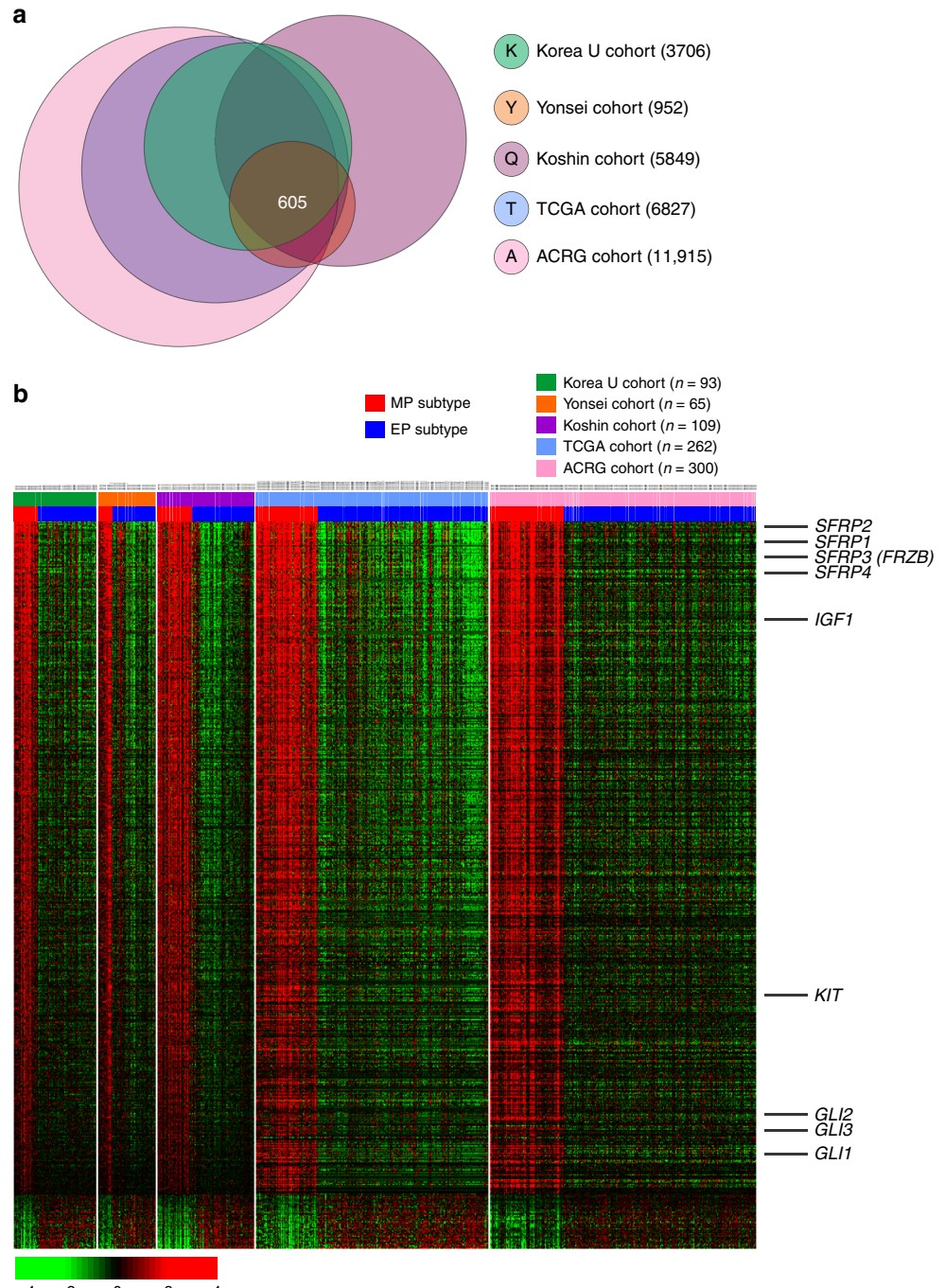

**Fig. 6** Subtype-specific gene expression patterns conserved in five cohorts of patients with gastric cancer. **a** Venn diagram of genes with expression that differed significantly between the mesenchymal phenotype (MP) and epithelial phenotype (EP) subtypes in five different cohorts (KUCM Kosin University College of Medicine, KUGH Korea University Guro Hospital, TCGA The Cancer Genome Atlas, YUSH Yonsei University Severance Hospital, ACRG Asian Cancer Research Group). Gene expression differences were considered statistically significant at $P < 0.001$ by Student's $t$-test. This stringent significance threshold was used to limit the number of false-positive findings. Expression of only 605 genes was upregulated or downregulated in all five cohorts. **b** Expression patterns of selected genes. The colored bars at the top of the heat map represent samples as indicated

with the demethylation agent 5-azacytidine (5-AzaC) restored expression of IGF1 in two of EP-like gastric cancer cells (MKN28 and SNU16, Supplementary Fig. 17). Furthermore, promoter regions of IGF1 were concomitantly demethylated upon treatment of 5-AzaC (Supplementary Fig. 18), supporting promoter hypomethylation as one of the mechanisms for activation of the IGF1/IGF1R pathway in gastric cancer. When MP signature-based BCCP model was applied to gene expression data from these cell lines, Hs746T and SNU1 were predicted as MP subtype

while MKN45 and SNU16 were predicted as EP subtype (Supplementary Fig. 19), further supporting molecular correlation between IGF1 expression and MP subtype in gastric cancer cells. In good agreement with our hypothesis, three MP-like cell lines with high IGF1 expression were significantly more sensitive to treatment of linsitinib in a dose-dependent manner (Fig. 8c). To explore this in a mouse model, we carried out a mouse xenograft experiment with SNU1 cells. Tumor growth was significantly delayed by linsitinib treatment (Fig. 8d), and tumor

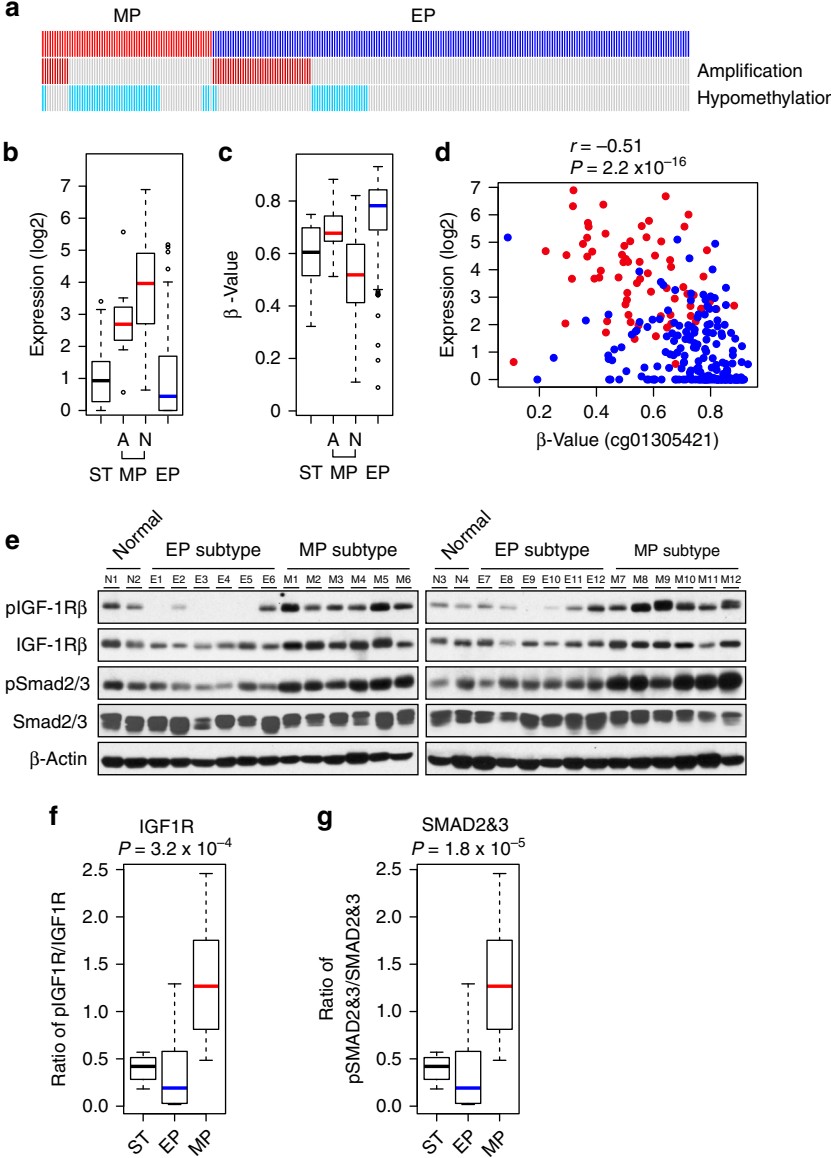

**Fig. 7** Activation of IGF1/IGF1R pathway in the mesenchymal phenotype (MP) subtype of gastric cancer. **a** Copy number amplification and hypomethylation status of the *IGF1* gene in MP subtype tumor samples ($n = 70$) from The Cancer Genome Atlas (TCGA) cohort. Hypomethylation of the promoter region was determined by comparison with methylation of surrounding normal tissues (<0.6). EP epithelial phenotype. **b** Expression of *IGF1* in MP subtype tumors, stratified by *IGF1* gene amplification in the cancer genome. A: amplification of IGF1, N: normal copy number, ST: surrounding tissue. Differences between ST and MP groups were significant ($P < 0.05$ by Student's $t$-test). Colored lines indicate the median, boxes extend from the 25th to the 75th percentile, and dashed error bars extend to the 10th and 90th percentiles. **c** Promoter methylation of the *IGF1* gene (probe ID, cg01305421) in each subtype of gastric cancer and ST in the TCGA tumor samples. Colored lines indicate the median, boxes extend from the 25th to the 75th percentile, and dashed error bars extend to the 10th and 90th percentiles. Differences between ST and MP groups were significant ($P < 0.05$ by Student's $t$-test). **d** Correlation between mRNA expression and promoter methylation (β-value), estimated using Pearson's correlation. **e** Western blot analysis for IGF1R, phospho-IGF1R, SMAD1/2, and phospho-SMAD2/3. Activation of IGF1R and SMAD2/3 was determined by western blot with phosphorylation-specific antibodies as indicated. β-Actin was used as loading control. **f**, **g** Activation of IGF1R and SMAD2/3 in gastric cancer and normal gastric tissues is estimated by ratios of phosphorylated form over all proteins. *P* values for mutation rates were obtained using Student's $t$-test (two-sided)

weight remained low (Fig. 8e, f). Consistently, growth of Hs746T tumors was also significantly reduced by linsitinib treatment (Supplementary Fig. 20). Notably, treatment of linsitinib significantly downregulated AKT and extracellular signal-regulated kinase (ERK) signaling (Supplementary Fig. 20d, e), the key downstream effectors of IGF1R. Together with data from cell lines, these results clearly demonstrate that MP gastric tumors are sensitive to inhibition of IGF1R.

**Similarity to other molecular subtypes**. Since MP subtype has clinical and molecular characteristics similar to previously recognized genomic diffuse (G-DIF) tumors in intrinsic subtypes[34], we compared two molecular subtypes after stratifying patients according to intrinsic genomic signature (G-DIF or genomic intestinal [G-INT] subtype). For convenience of comparison, we pooled three Illumina platform cohorts (KUGH, YUSH, and KUCM) and renamed them: KYK cohort. Consistent with a previous report[34], patients in the G-DIF subtype had a

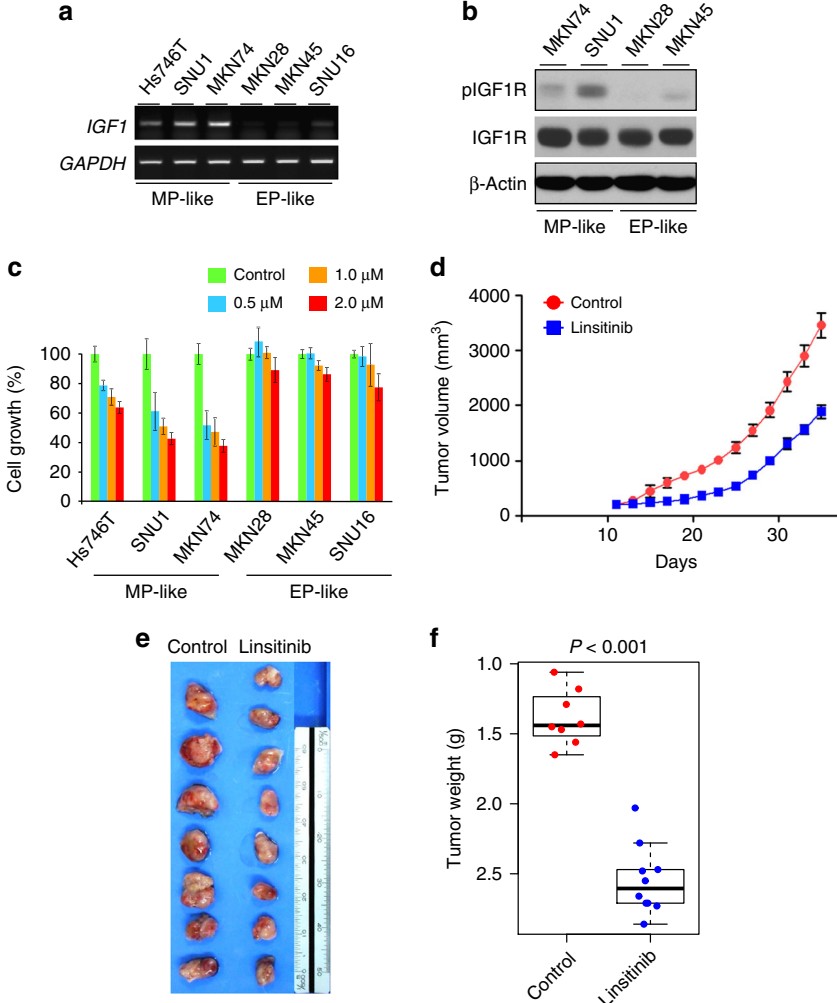

**Fig. 8** IGF1/IGF1R pathway is therapeutic target for MP subtype. **a** Expression of IGF1 in gastric cancer cell lines. Expression was measured by qRT-PCR. **b** Western blot analyses in gastric cancer cell lines of MP subtype cells (MKN74 and SNU1) and EP subtype cells (MKN28 and MKN45) using antibodies against phosphorylated IGF1R (active), IGF1R (total), and β-actin. **c** Sensitivity of gastric cancer cell lines to linsitinib, inhibitor of IGF1R. Cell growth was measured at 96 h after treatment of linsitinib as indicated ($n = 4$). **d** Growth of SNU1-derived xenograft tumors in mice treated with linsitinib or vehicle control. SNU1 cells were xenografted subcutaneously into the flanks of mice. At 10 days after xenografting, linsitinib or control tartaric acid was orally administrated to mice. Tumor volume was measured on the indicated days. Error bars indicate s.e.m. **e** Tumors harvested after treatment of linsitinib or control vehicle. **f** Tumor weight after treatment of linsitinib. At 25 days after linsitinib treatment, mice were killed and tumor weights were measured ($n = 8$ or 9 per treatment). Data are presented with means. $P$ values were obtained by Student's $t$-test

poor prognosis in all three cohorts as evidenced by significantly shorter RFS (Supplementary Fig. 21). A vast majority of patients in MP subtypes were G-DIF subtype, although it represents much larger patient population (Supplementary Fig. 21a), suggesting that MP subtype is a subset of G-DIF subtype. Interestingly, prognosis of patients in MP subtype were significantly worse than those in the remaining G-DIF subtypes in all three cohorts (Supplementary Fig. 21b), suggesting difference between MP and G-DIF subtypes. Furthermore, expression of IGF1, the most distinctive characteristic of MP subtype, is significantly higher in MP subtype than in G-DIF subtype (Supplementary Fig 22). These data suggested that MP subtype is biologically and clinically different from G-DIF subtype, although they share some similarity. Furthermore, only a few genes were shared by the two prognostic expression signatures (Supplementary Fig. 23). Interestingly, MP subtype also shared some clinical features with previously identified EMT subtype from ACRG study[19] as demonstrated by that EMT subtype is a subset of MP subtype and its survival pattern is very similar to MP subtype (Supplementary Fig. 24).

MP subtype is also similar to genomically stable (GS) subtype that was identified in a recent TCGA gastric cancer study[20]. In TCGA cohort, patients in MP subtype were divided into EBV ($n = 4$), MSI ($n = 6$), chromosomal instability ($n = 22$), and GS ($n = 38$) subtypes (Supplementary Table 5). Nonexclusive overlap between two subtypes indicate that MP subtype may be different from GS subtype although they share some molecular characteristics.

## Discussion

Using a series of independent but complementary approaches, we found that MP subtype gastric cancer has distinctive genetic and epigenetic alterations that are associated with poor prognosis and resistance to chemotherapy (Fig. 9). The existence of the MP subtype of gastric cancer is strongly supported by several lines of evidence in our study. First, the global gene expression patterns of

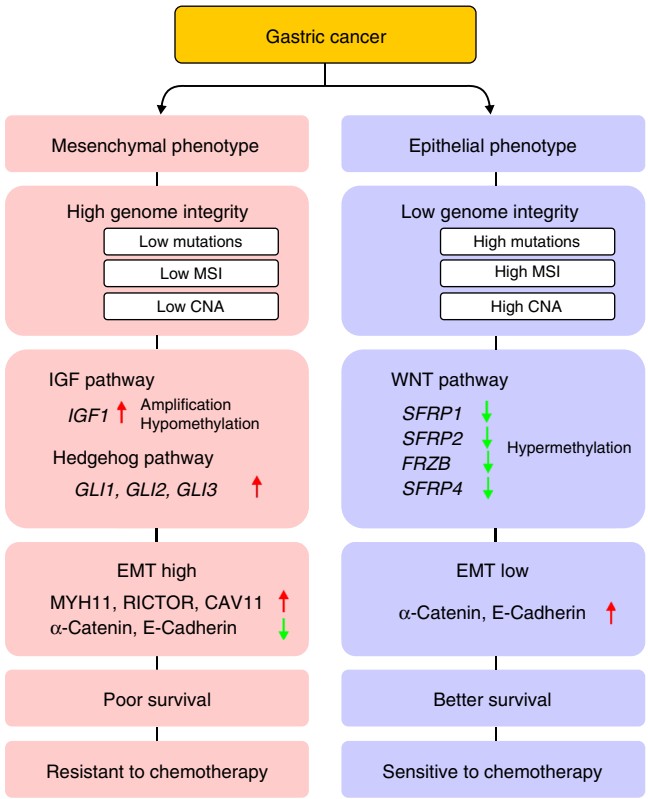

**Fig. 9** Summary of characteristics of the two subtypes of gastric cancer. CNA copy number alteration, EMT epithelial-to-mesenchymal transition, MSI microsatellite instability

MP subtype tumors were highly similar to those of GIST, a mesenchymal cancer. Second, signaling pathways that drive EMT, including the TGF-β pathway[12] and the hedgehog pathway[35], were highly activated in MP subtype tumors. Third, expression of E-cadherin and α-catenin was significantly reduced in MP subtype tumors. Fourth, expression of EMT-promoting proteins or miRNAs, including MYH11, RICTOR, CAV11, and miR-490, was significantly increased in MP subtype tumors. Lastly, recurrence rates reflecting clinical consequences of EMT[9–11] were significantly higher in patients with MP subtype tumors.

Our data clearly demonstrate the clinical significance of the two subtypes as tested and validated in six independent cohorts with a total of 1039 patients. In addition to the prognostic differences between the two subtypes, we found differences in response to standard treatments. Subset analysis of patients with available chemotherapy data strongly suggested that only EP subtype is associated with benefit from adjuvant chemotherapy. In patients with advanced disease, 5-fluorouracil–based chemotherapy was associated with improved outcomes for patients with EP subtype tumors, whereas this chemotherapy provided no benefit for those with MP subtype tumors.

Low sensitivity of MP subtype tumors to chemotherapy is consistent with our observation that MP subtype tumors are similar to GIST, which is associated with almost no benefit from adjuvant chemotherapy. GIST partial response rates to systemic chemotherapy have been shown to be less than 15%[36,37]. Previous studies have demonstrated a significant correlation between genome instability and increased sensitivity to cytotoxic chemotherapy in ovarian cancer, strongly suggesting that the high genomic instability in EP subtype tumors might contribute to their relatively higher sensitivity to adjuvant chemotherapy. Taken together, our results strongly indicate that the two

subtypes of gastric cancer are biologically distinct and might require different optimal clinical courses of treatment.

One of the most striking biological characteristics of EP subtype tumors was the concordant downregulation of all *SFRP*s, whose best-known function is inhibition of the WNT pathway[32], suggesting that the WNT pathway might contribute to the development and progression of EP subtype gastric tumors. Our integrated analysis clearly demonstrated that activation of the WNT pathway was mediated by hypermethylation of SFRP promoters. This observation is consistent with previous studies demonstrating that the WNT pathway is regulated by promoter methylation of SFRPs in gastric cancer cell lines[38–41] and provides further evidence that inactivation of SFRPs by promoter methylation is a clinically important epigenetic alteration in gastric cancer. These findings suggest that patients with EP subtype gastric cancer may benefit from treatment with DNA methylation inhibitors such as decitabine and azacitidine[40].

Our data also demonstrated that IGF1/IGF1R pathway is highly activated in MP subtype through demethylation of IGF1 promoter sequences. More importantly, MP subtype gastric cancer cells were more sensitive to inhibition of IGF1/IGF1R pathway. IGF1/IGF1R pathway has been considered as key therapeutic targets for many cancers[42,43]. Although more than 10 IGF/IGF1R inhibitors have entered clinical trials[44], many large clinical trials involving patients with non-small-cell lung cancer, breast cancer, and pancreatic cancer failed to show significant clinical benefit[45–47]. However, even among these unsuccessful trials, there are subsets of patients who have obtained benefit from IGF1R inhibition[48]. Since our data indicated that benefit of inhibiting IGF1/IGF1R pathway might be limited to MP subtype gastric cancer, possible reasons for current failure might be due to the lack of use of patient enrichment by biomarkers in the trials. Thus, our study may provide an opportunity to develop marker-based targeted therapy for gastric cancer which will increase patient benefit.

Our study has some limitations such as retrospective nature of clinical data. We cannot also rule out the possibility that cancer cells may induce expression of IGF1 from stromal cells by stimulating tumor microenvironment as tumors in MP subtype have higher fraction of non-tumor cells. Therefore, further in-depth examination of tumor microenvironment in both subtypes will be necessary in future studies.

In conclusion, we have identified two new subtypes of gastric cancer that are associated with significantly different survival outcomes. The MP genetic signature can be used to identify not only patients at high risk for recurrence (MP subtype), but also patients who would benefit from adjuvant chemotherapy (EP subtype) or targeted therapy (MP). Our data also make a case for targeted therapy as adjuvant therapy for MP gastric cancer patients. Further validation will be necessary before testing for the MP genetic signature can be implemented in routine clinical practice. Nevertheless, the validation of our findings in several independent patient cohorts and the fact that our two subtypes reflect biological characteristics associated with sensitivity or resistance to standard chemotherapy or targeted therapy indicates that our findings provide an opportunity to develop rational therapy recommendations. The association of the EP subtype with benefits from currently available standard treatments and the MP subtype with potential targeted treatments, if confirmed in prospective studies, could improve patient selection for these treatments.

## Methods
**Patients and samples**. We obtained fresh-frozen tumor specimens and clinical data from patients with gastric cancer who underwent gastrectomy as primary treatment at the KUGH (n = 93), YUSH (n = 65, or KUCM (n = 109), South

Korea, between 1999 and 2006. Frozen biopsy specimens of tumor tissues endoscopically collected before treatment from 40 patients with gastric cancer (treated from 2002 through 2010) were obtained from the fresh-frozen tissue bank of The University of Texas MD Anderson Cancer Center (the MDACC cohort). As an external validation set, we used additional tumor specimens collected from the SMC (n = 432) and ACRG (n = 300), as described in a previous study[18,19]. Clinical data of patients in all cohorts are available in Supplementary Data 2.

All samples were collected after written informed consent was obtained from patients, and the study was approved by the Institutional Review Board of The University of Texas MD Anderson Cancer Center (Houston, TX), KUGH (Seoul, Korea), YUSH (Seoul, Korea), KUCM (Busan, Korea), or SMC (Seoul, Korea). Clinical data were obtained retrospectively. OS was defined as the time from surgery to death, and RFS was defined as the time from surgery to the first confirmed recurrence. Data were censored when a patient was alive without recurrence at last contact. Of the 267 patients in the KUGH, YUSH, and KUCM cohorts, 155 had received standard adjuvant chemotherapy (either single-agent 5-fluorouracil or a combination of 5-fluorouracil and cisplatin/oxaliplatin, doxorubicin, or paclitaxel). All patients in the MDACC cohort received preoperative (neoadjuvant) chemotherapy or chemoradiation therapy. All patients in the SMC cohort received homogenous treatment with the INT-0116 regimen (5-fluoouracil/leucovorin and radiation) as adjuvant treatment[18]. Both OS and RFS data were available for patients in the KUGH, YUSH, KUCM, SMC, and ACRG cohorts, but only OS data were available for patients in the MDACC cohort.

Microarray experiments were performed with 267 samples of surgically removed frozen gastric cancer tissue (KUGH, YUSH, and KUCM cohorts), 40 samples of tumor biopsies (MDACC cohort), and 12 samples of surrounding non-tumor tissue. In addition, frozen samples of GIST tissue from the KUGH (three samples) were included in the microarray experiments. Patient characteristics for all cohorts are summarized in Supplementary Table 1. Data from the KUGH cohort were used to explore and identify potential prognostic signatures (exploration cohort, n = 93) and patient data from the remaining cohorts were used for validation of the prognostic signature (validation cohorts, n = 946)

**Gene expression data**. All of the experiments and analyses were conducted in the Department of Systems Biology at MD Anderson. Gene expression data from 307 patients with gastric cancer were generated by hybridizing labeled RNAs to human expression BeadChips (Human HT-12 v3.0; Illumina, San Diego, CA) containing 48,803 gene features. Briefly, total RNA was extracted from the fresh-frozen tissues using a mirVana RNA Isolation Labeling Kit (Ambion, Inc., Austin, TX). We used 500 ng of total RNA for labeling and hybridization, according to the manufacturer's protocols. Gene expression data from the SMC cohort were generated using HumanRef-8 WG-DASL v3.0 (Illumina) that contained a subset (24,526 gene features) of probes in Human HT-12. Gene expression data from the ACRG cohort were generated using Affymetrix Human Genome U133 Plus 2.0 Array. The microarray data were normalized using the quantile normalization method in the Linear Models for Microarray Data package in the R language environment[49]. The expression level of each gene was transformed into a log$_2$ base before further analysis. Primary microarray data are available in the Gene Expression Omnibus database of the National Center for Biotechnology Information (accession numbers GSE26899 for KUGH, GSE26901 for KUCM, GSE13861 for YUSH, GSE28541 for MDACC, GSE26253 for SMC, and GSE66229 for ACRG). Gene expression data from gastric cancer cell lines were obtained from GEO database (GSE22183).

**Genomic data from the TCGA cohort**. Genomic data from the TCGA gastric cancer cohort were downloaded from the TCGA data portal site (http://cancergenome.nih.gov/) and processed as described in previous studies[20,21,50–54]. Because mRNA expression data were available for 262 tumor tissues, all analysis with other data sets was limited to samples with available mRNA data. We analyzed 262 samples for methylation, 240 for miRNA, 260 for copy number alteration, and 257 for exome sequencing. Because most of the tissues in the TCGA cohort were recently collected, follow-up time of the patients in the TCGA cohort was very short and not sufficient enough for survival analysis. These patients were not included in survival analysis.

**Reverse phase protein arrays (RPPA)**. RPPA data were generated at MD Anderson. Briefly, protein was extracted using RPPA lysis buffer (1% Triton X-100, 50 nmol/L Hepes (pH 7.4)), 150 nmol/L NaCl, 1.5 nmol/L MgCl$_2$, 1 mmol/L EGTA, 100 mmol/L NaF, 10 mmol/L NaPPi, 10% glycerol, 1 nmol/L phenylmethylsulfonyl fluoride, 1 nmol/L Na$_3$VO$_4$, and 10 Ag/mL aprotinin) from human tumors and RPPA was performed as described previously[20,21,50,51]. Tumor lysates were adjusted to 1 μg/μL concentration, diluted in fivefold serial dilutions with lysis buffer, and printed on nitrocellulose-coated slides (Grace Bio-Labs, Bend, OR). Slides were probed with 191 validated primary antibodies followed by corresponding secondary antibodies (goat anti-rabbit IgG, goat anti-mouse IgG, or rabbit anti-goat IgG). Signal was captured using diaminobenzidine colorimetric reaction and normalized by using the fitted curve ("supercurve") approach[53]. In total, 255 samples were included in the analysis.

**Statistical analysis**. BRB-Array Tools (National Institutes of Health) was primarily used for all statistical analyses[55]. Cluster analysis was performed using Cluster and Treeview v3.0 (ref. [56]). We identified genes whose expression was unique to the S cluster (MP subtype) by cross-comparing gene lists from independent statistical tests. We first generated two different gene lists by applying 2-sample t-tests (P < 0.001). Gene list X represented genes that were differentially expressed between the S cluster (MP subtype) and L cluster (EP subtype), and gene list Y represented genes that were differentially expressed between the S cluster (MP subtype) and non-tumor gastric surrounding tissues from patients with gastric adenocarcinoma (Supplementary Fig. 1). By applying an additional threshold cutoff (twofold difference) to common genes in the two gene lists, we identified 299 genes whose expression patterns were specific to the S cluster (MP subtype) and that were potential markers for predicting prognosis. Multivariate Cox proportional hazards regression analysis was used to evaluate independent prognostic factors associated with survival, and gene signature, tumor stage, and pathologic characteristics were used as covariates. To assess the strength of the interaction between the two subtypes and adjuvant chemotherapy, a Cox proportional hazards model was fitted to data from patients in the pooled cohort. The model included three other covariates from the primary analysis (sex, age, and American Joint Committee on Cancer stage). All statistical analyses were conducted in the R language environment (http://www.r-project.org).

**Prediction models for validation of the MP signature**. Prediction of patient class in independent cohorts was done as described previously[57–62]. Briefly, gene expression data in the training set (KUGH cohort) were combined to form a classifier according to Bayesian compound covariate predictor (BCCP)[63], and the robustness of the classifier was estimated using a misclassification rate determined during the leave-one-out cross-validation (LOOCV) in the training set. When a classifier was applied to the independent validation sets (YUSH, KUCM, MDACC, SMC, and ACRG cohorts), prognostic significance was estimated using Kaplan–Meier plots and log-rank tests between two predicted subtypes of patients. After LOOCV, sensitivity and specificity of the prediction models were estimated using the fraction of samples correctly predicted. Specificity for correctly predicting the MP subtype by BCCP was 0.924, and sensitivity was 0.963. SVM algorithm was applied to gene expression data to test the robustness of the MP signature. Specificity and sensitivity of SVM predictor for MP subtype was 0.97 and 0.963, respectively. BCCP classifier was also applied to gene expression data from gastric cancer cell lines (GSE22183). Of the six cell lines used in drug sensitivity test, gene expression data were only available for four cell lines.

Previously defined G-IDF/G-INT gene expression signature (171 genes)[34] and BCCP algorithm were applied to gene expression data to stratify patients into G-DIF and G-INT subtypes.

**Selection of conserved MP-specific gene expression patterns**. Although the 299 gene expression signature was robust enough to identify patients with the MP and EP subtypes in all 6 cohorts (n = 1039 total), the number of genes was not sufficient enough to develop a sophisticated gene network analysis, because we applied an extremely stringent cutoff (P < 0.001 by Student's t-test and twofold difference) to avoid any potential false-positive results during signature-based prediction. Thus, to explore biological characteristics of the MP and EP subtypes, we selected genes whose expression patterns were conserved in the five cohorts (KUGH, YUSH, KUCM, TCGA, and ACRG). Gene expression data from the SMC cohort were not included because they were not generated using a full genome microarray and had only a limited number of genes. Likewise, gene expression data from the MDACC cohort were not included owing to small sample size. Gene lists K, Y, Q, T, and A (Fig. 6a) represented genes that were differentially expressed between the MP and EP subtypes in the KUGH, YUSH, KUCM, TCGA, ACRG cohorts, respectively. Expression of 605 gene features was significantly different between the MP and EP subtypes in all 5 cohorts.

**Bioinformatics analysis**. Ingenuity Pathways Analysis (Ingenuity Systems, www.ingenuity.com) was used for gene set enrichment analysis and gene network analysis. Gene set enrichment analysis was carried out to identify the most significant gene sets associated with disease process, molecular and cellular functions, and normal physiological and developmental conditions in selected genes as described in the instructions from the Ingenuity Systems. The significance of overrepresented gene sets was estimated using the right-tailed Fisher's exact test. Gene network analysis was carried out using a global molecular network developed from information contained in the Ingenuity Knowledge Base.

To estimate stromal content in tumor mass, we used analytical platform CIBERSORT (https://cibersort.stanford.edu/) that can quantify relative levels of the abundances of distinct cell types in a mixed cell population[31]. Mean gene expression data from tumors and surrounding tissues were used as reference (signature gene data) for tumor cell and normal gastric cells and individual tumor data were considered as mixture data as instructed. Analyses were done with 100 permutations with default statistical parameters. Estimated percentage of stromal cells in tumor mass ranged from 0 to 72%.

Histologically examined stromal cell percentage and CIBERSORT predicted stromal cell percentage were used to estimate contribution of stromal cells to gene

expression data from tumor mass. Contribution of stromal cells in gene expression data from tumor mass were removed by multiplying tumor cell percentages (0 to 1 scale) to gene expression data.

**Cell culture and 5-AzaC treatment.** Hs746T cell line was cultured in Dulbecco's modified Eagle's medium with high glucose and 4 mM glutamine (Invitrogen, Carlsbad, CA, USA). SNU1, MKN74, MKN28, MKN45, and SNU16 cell lines were cultured in RPMI 1640 (Invitrogen). All media were supplemented with heat-inactivated 10% fetal bovine serum and penicillin/streptomycin (Invitrogen). Gastric cell lines were treated with 2 μM of 5-AzaC for 72 h. No mycoplasma contamination was detected in any of the cultures using a mycoplasma detection kit. All cell lines were obtained from Yonsei University and authenticated using short tandem repeat, which was carried out at the Characterized Cell Line Core Facility, UT MD Anderson Cancer Center.

**Quantitative analysis of IGF1 and SFRP mRNA expression.** Total RNAs in 5-AzadC-treated or -untreated cells were extracted using TRIzol solution (Invitrogen, Carlsbad, CA), and then genomic DNAs were removed by DNAse I (Promega, Madison, WI, USA) treatment. Total RNAs (5 μg) were reverse-transcribed using a first-strand complementary DNA (cDNA) synthesis kit (Promega) according to the manufacturer's specifications. The cDNAs were subjected to quantitative polymerase chain reaction (qPCR) experiments using *IGF1, SFRP1, SFRP2, FRZB, SFRP4,* and *GAPDH* primer sets. The forward and reverse primers were 5′-GACTCTGAAACCTCAAGCTGTCT-3′ and 5′-GACA-GATGTAACGAATGGCCAGT-3′ for *IGF1,* 5′-CAATGCCACCGAAGCCTC-CAAG-3′ and 5′-CAAACTCGCTGGCACAGAGATG-3′ for *SFRP1,* 5′-CTCCAAAGGTATGTGAAGCCTGC-3′ and 5′-CCAGGATGATTTTGG-TATCTCGG-3′ for *SFRP2,* 5′-GCTACACAGAAGACCTATTTCCG-3′ and 5′-CCGTGGAATGTTTACCAGAGAGG-3′ for *FRZB,* 5′-CTAT-GACCGTGGCGTGTGCATT-3′ and 5′-GCTTAGGCGTTTACAGTCAACATC-3′ for *SFRP4,* and 5′-TTCGACAGTCAGCCGCATCTTCTT-3′ and 5′-CAGGCGCCCAATACGA CCAAATC-3′ for the GAPDH gene. Quantitative reverse transcription-PCR (qRT-PCR) was performed in triplicate on a Mastercycler ep realplex system (Eppendorf, Hauppauge, NY, USA). Cycling conditions were 95 °C for 30 s, followed by 40 cycles of 95 °C for 5 s, 62 °C for 30 s, and 72 °C for 20 s. Relative amounts of mRNAs were calculated from the threshold cycle number using expression levels of GAPDH as an endogenous control.

**Western blot analysis.** Tissue lysates were isolated using RIPA buffer (50 mM Tris-HCl pH 8.0, 150 mM NaCl, 2 mM EDTA, 0.5% Na-deoxycholate, 1% NP-40, 0.1% SDS) supplemented with protease inhibitors (Roche, Indianapolis, IN) and phosphatase inhibitors (Sigma). Then, 20 μg of the protein was separated by sodium dodecyl sulfate–polyacrylamide gel electrophoresis (SDS-PAGE) and transferred onto polyvinylidene difluoride membranes, were immunoblotted using antibodies against IGF-1Rβ, phospho-IGF-1Rβ, SMAD2/3, phospho-SMAD2/3 (Cell Signaling, Danvers, MA, Cat. No. 3027, 3918, 8685P, and 8828 respectively), or β-actin (Chemicon, Billerica, MA, Cat. No. MAB1501). Full scan of western blots are available in Supplementary Fig. 25. Antibodies were diluted in a ratio of 1:200. Proteins of interest were detected with horseradish peroxidase-conjugated sheep anti-mouse IgG antibody (GE Healthcare, Uppsala, Sweden) and visualized with the Pierce ECL Western blotting substrate (Thermo Scientific, Rockford, IL), according to the provided protocol.

**Cell viability assay.** Human gastric cell lines were cultured on 12-well plates at a density of 50,000 cells per well for 24 h and then treated with linsitinib, IGF-1R inhibitor (Selleck Chemicals, Houston, TX, USA), or equivalent volumes of vehicles to concentrations on exposure times indicated. The number of viable cells was counted by Trypan blue dye exclusion using the CountessTM automated cell counter (Invitrogen). The results were shown as a relative ratio to controls.

**Human gastric cancer cell line xenograft model.** To establish xenograft tumors, Hs746T or SNU1 cell lines ($1.9 \times 10^6$ cells/mouse) were injected subcutaneously into the upper left flank region of female BALB/c nude mice.

Experimental sample sizes were chosen using power calculations with preliminary experiments and/or were based on previously described variability in similar experiments.[64] After 10 days, tumor-bearing mice were grouped randomly ($n = 7$–9/group) and treated daily oral dosing at 60 mg/kg IGF-1R inhibitor, linsitinib (OSI-906). Tumor size was measured every other day using calipers. Tumor volume was estimated using the following formula: $L \times S2/2$ (where $L$ is the longest diameter and $S$ shortest diameter)[65]. Animals were maintained under specific pathogen-free conditions. All experiments were approved by the Animal Experiment Committee of Yonsei University.

**Immunohistochemistry and image analysis.** All tissues were fixed in 10% neutral-buffered formalin and embedded in paraffin wax using standard protocols. Tissue sections (5 μm) were dewaxed and antigen retrieval was performed in citrate buffer (pH 6), using an electric pressure cooker set at 120 °C for 5 min. Sections were incubated for 5 min in 3% hydrogen peroxide to quench endogenous tissue

peroxidase. Primary monoclonal antibodies directed against phospho-Akt1 (pS473) and phospho-ERK1/2 (pT202/pY204 for ERK1 and pT185/pY187 for ERK3) (ab8932 and ab50011, Abcam, Cambridge, UK) were diluted with phosphate-buffered saline in a ratio of 1:100. All tissue sections were counterstained with hematoxylin, dehydrated, and mounted. MetaMorph 4.6 software (Universal Imaging Co., Downingtown, PA, USA) was used for computerized quantification of immunostained target protein. The brown stain areas were identified, and the intensity was quantified using the software.

**Pyrosequencing.** Pyrosequencing was used to evaluate the methylation of *IGF1.* Briefly, 500 ng of total DNA from each of the gastric cancer tissue samples was used for bisulfite conversion using the EZ DNA Methylation Gold kit (Zymo Research, Orange, CA). Then, 1 μl of the bisulfite-converted DNA was used in a 20 μl PCR mixture containing primer sets and 2× Master Mix (Doctor Protein, Seoul, Korea) and amplified using a GeneAmp PCR system 9700 (Applied Biosystems, Waltham, MA). For pyrosequencing, forward, reverse, and sequencing primers were designed with PSQ Assay Design v1.0.6 (Biotage, Kungsgatan, Sweden). Standard pyrosequencing was then performed. Briefly, 20 μl of PCR product was immobilized on 3 μl of Streptavidin Sepharose High Performance (GE Health-care Bio-Sciences, Uppsala, Sweden) and annealed with sequencing primer for 10 min at 80 °C. Finally, the generated pyrograms were analyzed using PyroMark analysis software (Biotage). PCR condition and sequences for primer sets (Bioneer, Daejeon, Korea) are shown in Supplementary Table 6.

**Bisulfite sequencing.** For bisulfite treatment, 1 μg genomic DNAs were denatured in 0.2 mol/L NaOH. Sodium bisulfite (Sigma) and hydroquinone (Sigma) were added to final concentrations of 3.1 mol/L and 0.5 mmol/L, respectively, and the samples were incubated at 55 °C for 16 h. After purification using DNA Clean-Up System (Promega), the DNA samples were desulfonated in 0.3 mol/L NaOH, precipitated with ethanol, and resolved with 20 μL Tris-EDTA (TE) buffer. The modified DNAs (20 ng) were amplified by PCR using Taq DNA polymerase (New England Biolabs, Beverly, MA). Cycling conditions were 94 °C for 5 min, followed by 35 cycles of 94 °C for 30 s, 56 °C for 30 s, and 72 °C for 30 s, with a final extension of 5 min at 72 °C. The primer sequences for amplification of the *IGF1* promoter region were 5′-GTTTAGAAGAGGATTTTTATGGGT-3′ (sense, upper strand) and 5′-CCTAACAAAAATATATCTTTAACTCC-3′ (antisense, upper strand). The resulting PCR products were cloned into a pGEM-T-easy vector (Promega) and subjected to sequencing analysis. The primer sequences for bisulfite sequencing, which encompasses the probe region used in pyrosequencing (+130 to +143 nt. from transcription start site), were 5'-GTGTTTTGTAGA-TAAATGTGAGGA-3' (sense, upper strand) and 5'-CTACTAATTTTTCCCAT-TACTTCTA-3' (antisense, upper strand).

**Data availability.** The genomic data that support findings of this study are available from the NCBI Gene Expression Omnibus (GEO) under accession numbers GSE26899 for KUGH, GSE26901 for KUCM, GSE13861 for YUSH, GSE28541 for MDACC, GSE26253 for SMC, and GSE66229 for ACRG.

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

## Acknowledgements

This study was supported by Cancer Prevention & Research Institute of Texas (RP170307), Congressionally Directed Medical Research Programs (CA160616), National Institutes of Health grants (CA127672, CA129906, CA138671, CA150229, and

5U24CA143883), and IRG and SINF from The University of Texas MD Anderson Cancer Center. Additional support was provided by the National Institutes of Health through a Cancer Center Support Grant to MD Anderson (CA016672), the National Research Resource Bank Program of the Korea Science and Engineering Foundation in the Ministry of Science and Technology (Grant R21-2007-000-10058-0), the Korea Health Technology R&D Project through the Korean Health Industry Development Institute (No. H13C2162), National Research Foundation of Korea grant through MIST (No. 2017R1A2B2011684), and Research Initiative Grant from Korean Research Institute of Bioscience and Biotechnology.

## Author contributions

S.C.O., B.H.S., J.-H.C., and J.-S.L. designed the study, analyzed and interpreted the data, and wrote the manuscript. S.C.O., S.H.L., J.-H.C., Y.-J.J., Y.-J.M., W.J., B.-H.K., A.K., J.Y. C., J.Y.L., Y.H., S.S., E.E., J.S.E., J.H.L., M.S.B., S.H.N., J.L., W.K.K., S.K., and J.A.A. collected and processed tumor tissues. B.H.S., S.C.O., J.E.L., K.C.P., Y.-Y.P., Y.L., and W. L. performed experiments. S.-B.K., E.S.P., S.-C.K., and I.-S.C. performed informatical and statistical analysis. H.-S.L., H.-J.J., J.H., J.H.C., J.Y.C., J.Y.L., S.H.N., G.B.M., and J.A.A. provided critical analysis and reviewed the manuscript. All authors approved the final draft.

## Additional information

**Competing interests:** The authors declare no competing interests.

