## [Peer Review File · Nature Communications]

Reviewers' comments:

Reviewer #1 (Remarks to the Author):

This is a revised version of the manuscript previously submitted more than 1 year ago. In this revised version, the authors incorporated an additional validation set that was recently published by the ACRG (Nat. Med. 2015), which reported a molecular classification whereby a gene expression subtype known as the mesenchymal-like type, enriched in diffuse-type gastric cancer, was associated with worse prognosis. The authors have modified the classifier, incorporating the BCCP method, as well as providing comparison with SVM, to support the robustness of their classifier. The authors also used cybersort and a tumour purity parameter from the TCGA data to adjust for gene expression in an attempt to support their interpretation that the gene signature originated from tumor cells. Lastly, the authors performed additional functional assays using gastric cancer cell lines and generated a xenograft in one case in which there was a response to IGF1R inhibitor, Linsitinib.

The manuscript has improved compared with the previous version but there are still several critical issues the authors need to address:

1. There have been several publications suggesting a poor survival linked to the mesenchymal phenotype, including a paper from Patrick Tan's group reporting a TGF-beta associated module of stroma-related genes predicting poor prognosis in multiple datasets, with the signature correlating with the proportion of intra-tumoral stroma (Gut 2012, <http://dx.doi.org/10.1136/gutjnl-2011-301373>); another paper by Alex Boussioutas' group (Clinical Cancer Research 2014; 20:2761-72) involving a smaller set of patients, and the recent ACRG paper in Nat. Medicine 2015 (doi:10.1038/nm.3850). One paper has inferred that the signature actually originated from the stroma. Whilst in the previous version of the current paper, the authors attributed most of the gene signature to have arisen from the cancer cells, in this revised version, the authors attempted to perform additional normalization procedures to support the tumor-specific nature of the gene signature. However, it is unclear how this normalization has been done and whether this is the correct way to do it. In the methods section (page 30 line 708-711), the authors stated they used tumors and surrounding tissues as signature gene data for tumor cells and normal gastric cells respectively, whereas individual tumor data was considered as mixture data for assessment and estimation of the percentage of stromal cells in the tumor mass. In the rebuttal letter, pt 2, the authors state "By using gene expression data from surrounding tissues, we estimated percentage of stromal cells (or percentage of tumor cells) in each tumor mass." Surrounding tissue may contain mostly normal gastric epithelial cells (if only the mucosa was included), or a mixture of epithelial and stromal cells (if both the mucosa and the thick smooth muscle wall was included). In either circumstance, data from the surrounding tissue would not be a good source of gene expression for estimation of %stromal cells.

The authors then adjust the gene expression by multiplying tumor cell percentage to gene expression data. This is based on the assumption that all non-tumor content is from stromal cells, which has not taken into account tumor-to-tumor variation in other cell types, such as lymphoid cells, contaminating normal epithelial cells, etc.

> As correctly pointed out by reviewer, non-tumor surrounding tissues would be mixture of stromal cells and epithelial cells. Thus, they would not be ideal source for estimating stromal cell percentage. We used surrounding tissues as our reference tissues for stromal cells since it would be impossible to get only stromal cells from human normal stomach and generate gene expression data. However, since surrounding tissues do not contain any cancer cells, it would be still appropriate to use for estimating non-cancer cell percentages. We will not use term “stromal cell” but use “non-tumor cells” in revised manuscript in order to accurately reflect characteristics of tumors.

2. The authors claim all 4 SFRPs were down-regulated in EP subtypes and provided expression data from ST (surrounding tissue) of TCGA in Supp. Fig. 12. SFRP2 and SFRP4 showed elevated expression in MP subtypes compared with ST, SFRP4 showed elevated levels in EP compared with ST. The expression data could be explained by expression of these two SFRPs in stromal cells. Also, the variable expression levels in ST could be caused by variable amounts of inclusion of smooth muscle wall in the ST. The authors performed qRT-PCR in the YUSH cohort to support down-regulated expression of all 4 SFRPs (Supp. Fig. 15). Have the authors compared the qRT-PCR data with the corresponding gene expression data in the array analysis? What is the correlation between them? Is the gene expression data showing the same trend as the TCGA data? Also is the amount of stromal (smooth muscle wall) tissue known in the ST?

> As pointed out by reviewer, we cannot ignore the possibility that stromal cells contribute expression of SFRP in tumor mass. Or there might be another signaling circuit that regulates expression of SFRPs in gastric cancer. However, since methylation status is significantly associated with expression of SFRPs in TCGA data set, it would be fair to state that promoter methylation is at least part of regulatory mechanisms. Since our current study is limited to characterization of MP subtype, pursuing details of SFRP regulatory mechanism in EP subtype is really beyond the scope of current study. We currently plan to pursue this interesting regulatory mechanism on activation of WNT pathway in EP subtype. Expression data from qRT-PCR and microarray experiments in YUSH cohort are highly correlated as shown in new supplementary data (Supplementary Fig. 13).

3. Link to point 2, supplementary fig 15 and 18, primer sequences for qRT-PCR are provided for IGF1 only, but not for the SFRPs. The reverse primer of IGF1 has 6 nucleotide mismatches at its 5' end for unknown reasons. More importantly, blast search of the IGF1 qRT-PCR primer sequences showed that they reside in exon 4 and do not flank the intron. Thus, there is a high risk of amplifying contaminating DNA. Since the RNA extraction protocol was not provided in the methods, it is not certain whether DNase treatment was performed. Even if this was done, it may not be sufficient to remove all contaminating DNA. Given the discrepancies between the results from the TCGA expression array data for SFRP4 versus the qRT-PCR data, it would be crucial to re-examine the qRT-PCR protocol.

> We apologize for the lack of primer sequences of SFRP1, 2, 3 and 4 and the RNA extraction protocol. We now included them in the methods. Primer sequences of IGF1 in manuscript is not correct due to typo error. We corrected sequences in revised manuscript. We include DNase I treatment step in our qRT-PCR experiment protocol. Details of protocol is now included in revised manuscript.

4. Linking to the methylation analysis in relation to gene expression, if the expression of some of the SFRPs are contributed by a mixture of stromal and tumor cells, it would be difficult to infer methylation regulating its expression from pooled data. It is not clear from the methods section how the beta-value is derived from Fig. 5, Supp. Fig 12, 14, 18. There are many probes in the illumina array residing in different regions of the SFRPs and IGF1 gene, with very different methylation values, especially for those residing in the gene promoter versus the gene body. The probe(s) from which the methylation value and its relationship with gene expression were derived should be clearly stated. The authors need to demonstrate methylation in relation to expression and performance of 5-aza treatment for the other SFRPs in gastric cell lines in order to support the claim.

> Because methylation data from TCGA gastric cancer project were initially generated by using 27K arrays (illumina humanmethylation27) and later by using 450K, substantial number of samples has only methylation data on 27K probes. Thus, our analysis (as well as of TCGA team's) were limited to probes in 27K arrays. In humanmethylation27 array, there is only one methylation probe for IGF1 (probe id cg01305421). We included probe ID in revised manuscript. SFRP members has multiple probes that are located in CpG islands in 27K arrays. Most of probes showed significant negative correlation. We also included probe ID for each gene that were used in analysis (Fig. 7 and Supplementary Fig. 11). Because TCGA genomic data clearly demonstrated significant correlation as evidenced by all of methylation probes are negatively correlated with expression of SFRP family and SFRPs are not our major focus of current study, we did not carry out extra experiments.

5. The authors selected gastric cell lines expressing or not expressing IGF1 mRNA for 5-Aza treatment (Supp. fig. 20). They found re-expression of IGF1 after 5-aza treatment in 2 cell lines. In order to confirm that the different expression in these cell lines are indeed due to methylation, the authors should use pyrosequencing to demonstrate the methylation status of IGF1 (similar to what they did for gastric tumor tissues), and demethylation of the promoter following 5-Aza treatment.

> We carried out bisulfite sequencing of CpG sequences in promoter regions before and after 5-Aza treatment in cell line as suggested. IGF1 doesn't have traditional promoter sequence and highest CpG density area is located in next to Exon1 (known as P2 promoter). Previous study showed that P2 promoter is major site for epigenetic regulation of IGF1 expression (Clin Epigenetics. 2015 Mar 13;7:22). Sequencing data showed that methylation in CpG islands is removed after 5-Aza C treatment (Supplementary Fig. 19). Unfortunately, since this area is not covered by Illumina's 27K methylation array, we were not able to correlate our cell line data to tumors.

6. The authors have generated additional data to support activation of the IGF pathway by demonstrating higher levels of phosphorylation of IGF-1Rbeta in the MP subtype gastric tumor tissue through Western blotting. They then use gastric cell lines expressing or not expressing IGF1 mRNA for treatment with Linsitinib, and demonstrated a difference in response to growth inhibition. Since they have demonstrated a difference in IGF-1R beta phosphorylation level in MP gastric tissue, they should

also demonstrate the same difference in cell lines expressing or not expressing IGF1 mRNA (since no protein data is available) to provide further support of the existence of an autocrine loop.

> As suggested, we carried out western blot experiments and included them in revised manuscript. The data show that IGF1R is phosphorylated in IGF1 expressing cells (Supplementary Fig 17).

7. The authors classified gastric cancer cell lines into MP and EP subtypes using the same algorithm. They should provide data as to how many and which cell lines were assigned to each subtype, the gene expression level of the gene signature in these cell lines classified as MP, as these would provide stronger support for the epithelial source of these genes that can be produced by stromal cells.

> We included full prediction of cell lines in revised manuscript and gene expression patterns associated with two subtypes (Supplementary Fig 20).

8. The authors performed mouse xenograft experiments using one MP-like gastric cell line and showed a response to Linsitinib. However, experiments should be done in more than one cell line and should encompass both MP and EP groups to support a subtype-specific response. One can argue that the source of IGF1 could also be from stromal cells, such that even those EP cell lines not producing IGF1 may still benefit from Linsitinib treatment.

> We agree with the reviewer's comment. Since we have already provided data (Figure 8b) showing that EP-like gastric cell lines were not responsive to Linsitinib treatment using MTT assay, it is very unlikely that these cells would become sensitive to linsitinib in xenograft model. As correctly pointed out by reviewer, stromal cell may produce Igf1 when they interact with cancer cells. However, because stromal induced Igf1 is mouse protein, it may not be highly potent to human IGF1R. Furthermore, it is currently unknown if mouse Igf1 can activate human IGF1R as much as that human IGF1 does. Therefore, testing potential effects of stromal induced Igf1 would be inconclusive. Due to this limitation and uncertainty, we did not carried out xenograft experiments with EP cell lines.

9. The authors should reconsider their data in light of potential paracrine signaling versus autocrine signaling alone in a complex tumor environment in their experimental design, interpretation of results and discussion.

> We thank reviewer for providing critical viewpoints on our data. As suggested, we included potential limitation of our study and data in discussion.

10. The authors compared their classifier to G-Int and G-dif and claimed a superior performance. They should also compare their MP to the Mesenchymal-like type from the ACRG that has the worse prognosis. I suspect the results could be very similar.

> We included EMT subtype from ACRG study in ACRG cohort to assess concordance among three

classification approaches. It indicated that G-DIF is largest subtype, followed by MP subtype. EMT subtype is subset of MP subtype (Supplementary Fig 24).

11. Over a year has passed since submission of the current revision. As the clinical follow-up data of the key datasets are relatively short (a lot of cases less than 5 years), it would improve the power of the analysis if the follow-up data were updated.

> We updated clinical data and repeated all related analyses. With longer follow-up data, clinical association of two subtypes became more significant than previous analysis. For example, association of two subtypes with clinical outcomes in training set were more significant (P-values were changed from 0.05 and 0.04 to 0.007 and 0.006, Figure 1). And interaction of two subtypes with adjuvant chemotherapy was also more significant (P-value changed from 0.05 to 0.01).

Other specific comments:

12. Supp. Fig 15 - Note typo in the legend referring this to the TCGA cohort instead of the YUSH cohort.

> We fixed typos.

13. Supp. Fig. 22 - What do the grey boxes in the SMC cohort designate? Why is there such a small fraction of cases classified as MP in the SMC cohort?

> Gray boxes were error during conversion of graphics. It isn't clear why SMC cohort has smaller fraction of MP subtype. One possibility might be that intestinal subtype in SMC cohort is smaller than other cohorts as intestinal type is loosely associated with MP subtype.

Reviewer #2 (Remarks to the Author):

This is the revision from Oh et al.:

The revision reads very well and the authors have done a great job in validating the MP (mesenchymal phenotype). My main issue is novelty. 4 papers have been published to illustrate this entity:

1) Gastroenterology. 2011 Aug;141(2):476-85, 485.e1-11. doi: 10.1053/j.gastro.2011.04.042. Epub 2011 Apr 28.

Intrinsic subtypes of gastric cancer, based on gene expression pattern, predict survival and respond differently to chemotherapy.

Tan IB1, Ivanova T, Lim KH, Ong CW, Deng N, Lee J, Tan SH, Wu J, Lee MH, Ooi CH, Rha SY, Wong WK, Boussioutas A, Yeoh KG, So J, Yong WP, Tsuburaya A, Grabsch H, Toh HC, Rozen S, Cheong JH, Noh SH,

Wan WK, Ajani JA, Lee JS, Tellez MS, Tan P.

2) Gastroenterology. 2013 Sep;145(3):554-65. doi: 10.1053/j.gastro.2013.05.010. Epub 2013 May 14. Identification of molecular subtypes of gastric cancer with different responses to PI3-kinase inhibitors and 5-fluorouracil.

Lei Z1, Tan IB, Das K, Deng N, Zouridis H, Pattison S, Chua C, Feng Z, Guan YK, Ooi CH, Ivanova T, Zhang S, Lee M, Wu J, Ngo A, Manesh S, Tan E, Teh BT, So JB, Goh LK, Boussioutas A, Lim TK, Flotow H, Tan P, Rozen SG.

3) Comprehensive molecular characterization of gastric adenocarcinoma.

Cancer Genome Atlas Research Network.

Nature. 2014 Sep 11;513(7517):202-9. doi: 10.1038/nature13480. Epub 2014 Jul 23.

4) Molecular analysis of gastric cancer identifies subtypes associated with distinct clinical outcomes.

Cristescu R, Lee J, Nebozhyn M, Kim KM, Ting JC, Wong SS, Liu J, Yue YG, Wang J, Yu K, Ye XS, Do IG, Liu S, Gong L, Fu J, Jin JG, Choi MG, Sohn TS, Lee JH, Bae JM, Kim ST, Park SH, Sohn I, Jung SH, Tan P, Chen R, Hardwick J, Kang WK, Ayers M, Hongyue D, Reinhard C, Loboda A, Kim S, Aggarwal A.

Nat Med. 2015 May;21(5):449-56. doi: 10.1038/nm.3850. Epub 2015 Apr 20.

and also this most recent paper :

Stromal-Based Signatures for the Classification of Gastric Cancer.

Uhlik MT, Liu J, Falcon BL, Iyer S, Stewart J, Celikkaya H, O'Mahony M, Sevinsky C, Lowes C, Douglass L, Jeffries C, Bodenmiller D, Chintharlapalli S, Fischl A, Gerald D, Xue Q, Lee JY, Santamaria-Pang A, Al-Kofahi Y, Sui Y, Desai K, Doman T, Aggarwal A, Carter JH, Pytowski B, Jaminet SC, Ginty F, Nasir A, Nagy JA, Dvorak HF, Benjamin LE.

Cancer Res. 2016 May 1;76(9):2573-86. doi: 10.1158/0008-5472.CAN-16-0022.

The only novelty in this study is the IGF1 axis and the therapeutic implication. The authors did cell lines studies and one xenograft, if this part of the study carries the main weight due to lack of novelty, then more functional studies should be carried out such as overexpression of IGF1 in the IGF1-low cell lines and vice versa (stable knock-down of overexpressed lines) then subject to IGF1 inhibitors to see the effects accordingly.

> As pointed by reviewer, our study shares some of findings with previous studies. However, our new subtypes are not identical to previous subtypes (while shared some features) and we uncovered difference in underlying biology between two subtypes. Furthermore, our findings were validated in multiple larger independent cohorts. Therefore, we believe that our new findings significantly improved

our understanding on development of gastric cancer in molecular level and provided therapeutic opportunity for poor prognostic patients by targeting IGF1R pathway with currently available drugs.

Reviewer #3 (Remarks to the Author):

The authors have demonstrated that gastric cancer samples divide into two molecular subtypes, based on mutations, epigenetics and gene expression. This stratification identifies a poor prognosis subtype, more effectively than previous histologic or molecular subtypes.

It is my impression that the authors have satisfied all the request of the reviewers with their recent changes. There are two minor changes I request to perfect the manuscript.

1) on Figure 5, the amplification status and hypomethylation of the IGF1 promoter needs to be demonstrated on EP cases as well as the MP cases. Otherwise it is hard to assess the contribution of this control mechanism in the two subtypes of the disease, EP/MP.

> We now included all of data in figure 5 (Figure 7 in revised manuscript).

2) A major concern about the presented evidence for molecular differences that make the group division between EP and MP is that the parameter that separates them is really continuous and not quantile. In that context, the conclusion that MP cases carry less mutations and genomic aberrations, is confounded by the presence of more stroma in those sections, introducing more wild type DNA. To negate that concern, I suggest that the supplementary figure 7 should actually be in the manuscript body, because in my impression, it shows convincingly that the authors do see mutations in MP cases when they're there.

> We now included supplementary Fig 7 to main body.

Reviewers' comments:

Reviewer #1 (Remarks to the Author):

In the current version, the authors have provided more details with clarification of data, performed minor change in wording, description or discussion compared with the previous version as well as updating the survival data encompassing longer follow up period. The only additional experiments provided include detection of IGF1R phosphorylation in gastric cell lines and analysis for demethylation after 5-Aza treatment in one gastric cancer cell line.

Specific comments:

Supplementary Fig. 17. It is not certain why IGF1R phosphorylation status is performed in only 4 out of the 6 cell lines. Specifically, data from the cell line Hs746, in which xenograft with Linsitinub treatment has been performed, is not available.

Supplementary Fig. 19. The authors performed clonal bisulphite sequencing in a gastric cancer cell line interrogating a different promoter region compared to the Illumina 27K methylation probe that display an association with IGF1 expression in tumor tissue. What is the status of methylation of the specific CpG site associated with the Illumina 27K probe in these cell lines then?

Other minor comments:

Page 12, first sentence "Down-regulation of SFRPs in MP subtype was further validated in another cohort (YUSH, Supplementary Fig. 12 and 13)."

Should it be EP subtype?

Supplementary Fig 10 legend : Adjustment should be for "non-tumor cells" instead of "stromal cells or stromal effect" to be consistent with the manuscript text.

Reviewer #1

In the current version, the authors have provided more details with clarification of data, performed minor change in wording, description or discussion compared with the previous version as well as updating the survival data encompassing longer follow up period. The only additional experiments provided include detection of IGF1R phosphorylation in gastric cell lines and analysis for demethylation after 5-Aza treatment in one gastric cancer cell line.

Specific comments:

Supplementary Fig. 17. It is not certain why IGF1R phosphorylation status is performed in only 4 out of the 6 cell lines. Specifically, data from the cell line Hs746, in which xenograft with Linsitinub treatment has been performed, is not available.

>As suggested, we included new western blots with Hs746T (new Supplementary Fig. 17). While p-IGF1R level is higher than two EP-like cells, it was lower than other MP-like cells. Thus, to ensure reproducibility of therapeutic effect of IGF1R inhibition in MP-like cells, we repeated experiments with another MP-like cell SNU1. Consistently growth of SNU1 xenograft tumors was significantly reduced by linsitinib treatment. Since SNU1 is better model for MP subtype, we included new data with SNU1 in main figure and moved old data with Hs746T to supplementary data.

Supplementary Fig. 19. The authors performed clonal bisulphite sequencing in a gastric cancer cell line interrogating a different promoter region compared to the Illumina 27K methylation probe that display an association with IGF1 expression in tumor tissue. What is the status of methylation of the specific CpG site associated with the Illumina 27K probe in these cell lines then?

>As suggested, we now included CpG site in 27K methylation array, probe id:cg01305421 (Supplementary Fig. 19b). Consistent with other CpG sites, it is methylated in EP-like cells and demethylated by treatment of 5AzadC.

Other minor comments:

Page 12, first sentence "Down-regulation of SFRPs in MP subtype was further validated in another cohort (YUSH, Supplementary Fig. 12 and 13)."

Should it be EP subtype?

>Yes, it should be EP. Corrected in the revised manuscript.

Supplementary Fig 10 legend: Adjustment should be for "non-tumor cells" instead of "stromal cells or stromal effect" to be consistent with the manuscript text.

>Corrected as suggested.

REVIEWERS' COMMENTS:

Reviewer #1 (Remarks to the Author):

In the current version, the authors have performed xenograft experiment with linsitinib treatment in an additional MP cell line SNU1 and demonstrated consistent therapeutic response as Hs746T. They have also performed additional experiment of clonal bisulphite sequencing covering the methylation array probe cg01305421 with consistent results. These additional studies help to strengthen their hypothesis.

Response to reviewer's comments

In the current version, the authors have performed xenograft experiment with linsitinib treatment in an additional MP cell line SNU1 and demonstrated consistent therapeutic response as Hs746T. They have also performed additional experiment of clonal bisulphite sequencing covering the methylation array probe cg01305421 with consistent results.

These additional studies help to strengthen their hypothesis.

> We found that there is no critique on our revised manuscript.